# Bioremoval of Methylene Blue from Aqueous Solutions by Green Algae (*Bracteacoccus* sp.) Isolated from North Jordan: Optimization, Kinetic, and Isotherm Studies



**Abdullah T. Al-Fawwaz [1],*** , **Ahmad Al Shra'ah [2] and Engy Elhaddad [3]**

1 Department of Biological Sciences, Al al-Bayt University, P.O. Box 130040, Al-Mafraq 25113, Jordan
2 Department of Chemistry, Faculty of Science, Al al-Bayt University, P.O. Box 130040, Al-Mafraq 25113, Jordan
3 National Institute of Oceanography and Fisheries (NIOF), Cairo 1981505, Egypt
* Correspondence: al_fawwaz@aabu.edu.jo; Tel.: +962-7-7762-6144

**Abstract:** Algae provide an alternative, sustainable, and environmentally beneficial method of dyetreatment. In this study, algae were successfully used to remove methylene blue (MB) from aqueous solutions. The effects of several parameters, such as initial concentration of MB (5–25 mg L$^{-1}$), algae dosage (0.02–0.1 g mL$^{-1}$), temperature (4, 20, and 30 °C), and contact time (24, 48, 72 and 84 h), on MB removal were investigated. In addition, the characterization of MB before and after treatment was achieved using UV-spectrophotometer and Liquid chromatography-mass spectrometry (LC-MS). The experimental data were applied to three kinetic models, namely pseudo-first-order, pseudo-second-order, and Elvoich. Moreover, Langmuir, Freundlich, Dubinin–Raduskevich (D–R), and Temkin isotherm models were tested. The maximum removal efficiency of MB (~96%) was accomplished at optimum conditions at the initial concentration of MB (15 mg L$^{-1}$), temperature (30 °C), and algae dosage (0.06 g mL$^{-1}$) after 60 min of contact time. The removal of MB follows the pseudo-second-order kinetic model (R$^2$ > 0.999), and the experimental data is best fitted by the Langmuir isotherm model (R$^2$ > 0.9300).

**Keywords:** algae; methylene blue; bioremoval; kinetics; equilibrium studies

## 1. Introduction

The industrial effluents from the textile, cosmetic, paper and printing, rubber, leather, pharmaceutical, food, and plastic industries contain many toxic pollutants, such as dyes, which harm the environment [1]. Dyes can be classified into two categories: non-ionic (vat and disperse dyes) and ionic (cationic (basic) and anionic dyes (reactive, direct, and acidic)) dyes [2]. The presence of dyes in aquatic ecosystems reduces light penetration and thus decreases the photosynthesis necessary for living organisms [3]. In addition, industrial dyes are mutagenic, toxic, and dangerous to humans and animals [4]. Therefore, the removal of dyes from aquatic environments is a necessary process to prevent water pollution [5]. Several processes, including mechanical (e.g., filtration and reverse osmosis), physical (e.g., adsorption, extraction, and flocculation), chemical (e.g., precipitation, oxidation, ion exchange, and ozonolysis), and thermal (e.g., evaporation and distillation) methods, have been used to remediate the contaminated water by dyes [6]. However, these methods have clear disadvantages, such as high cost, secondary pollution, not being effective on a large scale (e.g., remediation of polluted soil), and low efficiency [7]. Therefore, bioremediation using microbial species is a favored alternative process for being highly effective with low cost and environmentally friendly [8].

Algae contain main groups, such as polysaccharides, polyunsaturated fatty acids, pigments, proteins, lipids, vitamins, phycobiliproteins, and enzymes [9]. Consequently, the bio-removal of dyes by algae occurs via the interactions between the functional groups in algae (e.g., carboxylate, hydroxyl, sulfate, phosphate, and amino) with dye molecules [10].

Algae are highly available, and their growth is easy and fast [11]; thus, algae are widely used in environmental remediation for wastewater decolorization [12]. For example, the decolorization of tartrazine by *Scenedesmus bijugatus* (green algae) [13], direct brown NM dye by immobilized *Chlorellaalgae* [14], methyl red, orange II, G-Red (FN-3G), basic cationic, and basic fuchsin by *Chlorella vulgaris*, *Lyngbyalagerlerimi*, *Nostoclincki*, *Oscillatoria rubescens*, *Elkatothrixviridis* and *Volvox aureus* [15] were successfully achieved in previous studies. In addition, malachite green was highly removed by *Chlorella* algae [16] and freshwater algae *Pithophora* sp. [17].

Methylene blue is a common dye that is largely utilized in industry. Dye leftovers can be found in effluent discharged by such enterprises. As a result, the presence of very low quantities in the effluent is readily apparent. The discharge of methylene blue can produce a variety of issues, including increased COD by the water body, increased toxicity, decreased light penetration and photosynthesis, and damage to the aesthetic character of the water surface. Furthermore, the products of their degradation may be mutagenic and carcinogenic.

The ability of green algae of various types to break down or decolorize large concentrations of poisonous azo dyesis accomplished through many techniques.

Much researchremains to be completed in order to fully comprehend the entire mechanism(s) of deterioration.

To the best of our knowledge, this study is the first to optimize experimental parameters for removing MB from an aqueous solution using *Bracteacoccus* sp. isolated from northern Jordan and attempt to explain the removal mechanism through kinetic and equilibrium studies. Consequently, the main objectives of the current study were to estimate the ability of *Bracteacoccus* sp.to remove MB from an aqueous solution where MB dye was selected as a model compound of cationic days, to determine the optimum conditions for obtaining the highest removal of MB, and to illustrate the mechanism for MB removal by applying the kinetic and the equilibrium isotherm models to the experimental data.

## 2. Materials and Methods

### 2.1. Materials

All chemicals, such as methylene blue, which is used in the current study, are of analytical grade and used as received without further purification.

The MB used in this study was chosen as the model compound of cationic dyes, and its properties are listed in Table 1. A stock solution of MB 1000 mg $L^{-1}$ was prepared by dissolving 1.000 g of MB in 50 mL of deionized water and then diluted to 1000 mL. This stock solution was used to prepare the diluted solutions of MB in the range from 5 to 25 mg $L^{-1}$.

**Table 1.** Properties of methylene blue (MB) dye [18].

| Chemical Structure | |
|---|---|
| Molecular formula | $C_{16}H_{18}N_3ClS$ |
| Molecular weight | 319.85 g $mol^{-1}$ |
| Density | 1.1 g $cm^{-3}$ |
| The length of MB molecule | 13.82 Å or 14.47 Å |
| The width of MB | 9.5 Å |
| pKa | 3.8 |
| Solubility in water | 43.6 g $L^{-1}$ at 25 °C |
| Melting point | 100–110 °C |
| $\lambda_{max}$ | 664 |

### 2.2. Microalgal Culture

Green microalgae were collected from a water stream originating from a water spring located west of Mafraq city in north Jordan (location: 32°12′17.6″ N 36°00′05.1″ E). The samples were directly transferred to the laboratory.

### 2.3. Isolation, Cultivation, and Classification of Microalgae

Green microalgal samples were cultivated in flasks containing Bold Basal Medium (BBM). The isolation of algal colonies was conducted by a series of subcultures on BBM agar plates. Once algal colonies were separated, a pure culture was prepared and microscopically examined. Routine cultivation was conducted at $25 \pm 2$ °C under a light intensity of 20.25 $\mu Em^{-2}S^{-1}$ for 25 days. A pure culture was chosen and was identified as *Bracteacoccus* sp.

### 2.4. Characterization of Dye Aqueous Solutions

The treated and control samples of MB were characterized using the following methods. (a) UV–vis spectroscopy (Specord S 600-Molecular Spectroscopy-UV Vis Diode-array Spectrophotometers, Germany). In this method, the treated and control samples of MB were measured by a UV-visible spectrum at a scanning speed of 210–790 nm with a quartz cuvette. (b)High-resolution TOF-MS/MS (impact II-Bruker) (A Bruker Daltonik (Bremen, Germany) Impact II ESI-Q-TOF System equipped with Bruker Dalotonik Elute UPLC system (Bremen, Germany) was used for screening compounds of the treated and control samples of MB.

a.    Immobilization of *Bracteacoccus* sp.

Immobilization of *Bracteacoccus* sp. cells in a sodium alginate (Na-alginate) solution was prepared by the method described by [14]. A total of 3% of the Na-alginate solution was mixed with the algal suspension to give a 2:1 ratio (alginate to algal suspension (*v/v*)). To create blank beads (alginate without algae), 3% sodium alginate was mixed with deionized water. The 3% alginate–algae mixture, or alginate suspension, was then introduced dropwise (8–10 cm away from CaCl₂ solution) into 0.03 M CaCl₂ to produce the beads. The beads started to harden almost immediately when dropped into the CaCl₂ solution and were left in the solution to fully harden. The beads were then washed three times in deionized water and stored at 4 °C until used.

b.    Batch experiments

The effects of the initial MB concentration, temperature, immobilization, algae dosage, and contact time on the removal of MB from the aqueous solution were investigated. In detail, fresh algae biomass (equivalent to 0.03 g mL$^{-1}$ dry weight) was added to 20 mL of different MB solution (5–25 mg L$^{-1}$) in a 50 mL glass test tube, and then the batch temperature was adjusted to 4, 20, and 30 °C. The initial pH of the MB solution and the initial pH of the MB with the algae mixture was measured, and it was approximately constant (~7), which was reported in the literature as the favored pH value for obtaining the highest removal efficiency (>94%) [19].

Subsequently, the test tube with the MB and algae mixture was kept in the dark for different time intervals (24, 48, 72, and 84 h). After a specific interval of time, the mixture was centrifuged at 3500 rpm for 10 min. Then the absorbance of the cell-free supernatant was measured at the maximum absorption wavelength ($\lambda$ = 664 nm) using a UV spectrophotometer. A calibration curve of standard MB solutions in the range from 2 to 30 mg L$^{-1}$was also prepared with a good correlation factor ($R^2$ = 0.998). All experiments in this study were conducted in triplicate. The removal percentage (R%) of MB and the removed quantity of MB from the aqueous solution were calculated using Equations (1) and (2), respectively [20].

$$R\% = \frac{Co - Ct}{Co} \times 100 \tag{1}$$

$$qt = \frac{(Co - Ct)}{W} \times V \tag{2}$$

where $C_o$ and $C_t$ are the initial concentration of MB and the MB concentration at time t, respectively. The terms $q_t$ is the quantity of MB removed in mg per gram of algae, V is the volume in L of the solution, and W is the mass of algae in grams. In immobilization experiments, the difference between blank beads and beads with microalgae in dye concentration was considered.

c.　　Removal Kinetics

To understand the mechanism of MB removal using algae, three kinetic models, namely the pseudo-first-order [21], pseudo-second-order [22], and Elovich [23] models were tested. These models are expressed in Equations (3)–(5), respectively:

$$\ln(q_e - q_t) = \ln q_e - tK_1 \tag{3}$$

$$\frac{t}{q_t} = \frac{1}{K_2 q_e{}^2} + \frac{t}{q_t} \tag{4}$$

$$q_t = \frac{1}{\beta} \ln(\alpha.\beta) + \frac{1}{\beta} \ln t \tag{5}$$

where $q_e$ (mg g$^{-1}$) is the amount of MB adsorbed at equilibrium, $q_t$ (mg g$^{-1}$) is the amount of MB adsorbed at time t (min), and $k_1$ is the rate constant (min$^{-1}$). The linear plot between ln ($q_e - q_t$) versus t can be used to calculate the $k_1$ (slope) and $q_e$ from intercept (ln$q_e$). $k_2$ is the rate constant of pseudo-second-order adsorption (g mg$^{-1}$ min$^{-1}$), $\beta$ is the extent of surface coverage (min g mg$^{-1}$), $\alpha$ is the rate of adsorption (mg g$^{-1}$ min$^{-1}$), and t is time (5–360 min).

d.　　Removal isotherms

The isotherm adsorption equilibrium was studied using three isotherm models, namely the Langmuir [24], Freundlich [25], Temkin [26], and Dubinin–Raduskevich (D–R) [27] models. According to the Langmuir isotherm model, a monolayer of adsorbate molecules is formed on the homogenous adsorbent surface; therefore, the adsorption occurs on specific equal sites available onto adsorbate while ignoring the side interactions and steric hindrance. The equation of the Langmuir isotherm model is given in Equation (6):

$$\frac{Ce}{q_e} = \frac{1}{q_{max}} KL + \frac{1}{q_{max}} Ce \tag{6}$$

where $C_e$ is MB concentration at equilibrium (mg L$^{-1}$), $q_{max}$ is the maximum adsorption capacity (mg g$^{-1}$), and $K_L$ is the Langmuir constant. The nature of MB adsorption can be predicted by the term $R_L$, which represents a significant characteristic of the Langmuir model and can be calculated from Equation (7):

$$RL = \frac{1}{1 + KLCo} \tag{7}$$

where $C_o$ is the initial MB concentration (mg L$^{-1}$). $R_L$ is favored if its value is located between 0 and 1. Based on $R_L$ values, equilibrium is linear at $R_L = 1$, unfavored at $R_L > 1$, and irreversible at $R_L = 0$. The plot of $C_e/q_e$ versus $C_e$ gives a straight line with the slope $1/q_{max}$ and the intercept $(1/q_{max}) K_L$.

In addition, the adsorption process onto heterogeneous surfaces forms multilayers and can be described by the Freundlich adsorption isotherm model. The Freundlich adsorption isotherm model is given in Equation (8):

$$\log q_e = \log Q_f + \frac{1}{n} 1nL\log C_e \tag{8}$$

The plot of log $q_e$ versus log $C_e$ gives a straight line with slope $1/n$ (the heterogeneity factor) and intercept log $Q_f$, where $Q_f$ is the Freundlich constant. The Temkin isotherm model explains the interactions between adsorbate and adsorbent and determines whether the adsorption process is physisorption or chemisorption. The Temkin isotherm model is given in Equation (9):

$$q_e = \beta_1 \ln K_T + \beta_1 \ln C_e \tag{9}$$

where $K_T$ is the equilibrium constant for binding energy (L mol$^{-1}$), $\beta_1$ is a constant related to adsorption heat ($\beta_1 = RT/b$), R is the universal gas constant (8.314 J K$^{-1}$ mol$^{-1}$), and T is the temperature (K). Based on the b value, the sorption can be physisorption (8.0 kJ mol$^{-1}$) or chemisorption (between 8.0 and 16.0 kJ mol$^{-1}$). The plot $q_e$ versus $\ln C_e$ gives a straight line with the slope $\beta_1$ and the intercept $\beta_1 \ln K_T$.

Dubinin–Raduskevich (D–R) isotherm model is used to determine whether adsorption is physisorption or chemisorption. The D–R isotherm model is expressed by Equation (10):

$$\ln q_e = \ln q_m - K_D \varepsilon^2 \tag{10}$$

where qm is the theoretical isotherm saturation capacity (mg g$^{-1}$), KD is constant related to the mean free adsorption energy (mol$^2$·kJ$^{-2}$), $\varepsilon$ is the Polanyi potential (KJ mol$^{-1}$), and it is expressed in Equation (11):

$$\varepsilon = RT \ln\left(1 + \frac{1}{C_o}\right) \tag{11}$$

where R is the universal gas constant (8.314 × 10$^{-3}$ kJ mol$^{-1}$K$^{-1}$), and T is the temperature (K). The mean free energy of sorption E (KJ mol$^{-1}$) represents the change in Gibbs free energy per mole ion transferred to the surface of the solid from the solution, which is equal to $1/\sqrt{2KD}$. The plot of $\ln q_e$ versus $\varepsilon^2$ gives a straight line with the slope K and the intercept $\ln q_D$.

## 3. Results and Discussion

### 3.1. Characterization of Dye Aqueous Solutions

At the end of time terms, the adsorbent was removed by centrifugation and the supernatant was analyzed by UV-Visible spectroscopy for the residual of MB with a measurement range from 200 to 800 nm and a data interval of 10.0 nm (Figure 1). The UV-visible spectra of MB before and after treatment with *Bracteacoccus* sp. were determined after 84 h of exposure; the results in Figure 1 display the UV-visible spectra of MB before and after treatment with algal biomass. $\lambda_{max}$ was found to be around 664 nm before treatment, and the spectra were different. Some peaks disappeared after treatment, which means that the MB was removed from the solution, and this can be confirmed by visual observation, as shown in Figure 2.

As shown in Figure 1, methylene blue before treatment has two absorbance peaks at 610 and 670 nm; the peak at 610 nm is not a band but a shoulder, and it appears because of a transition between the two levels: the ground state and the excited state. The two peaks disappeared after treatment with algae. The reason behind color disappearance is the binding process that occurs on the surface of algal biomass. The sorption of MB into algae may occur through the interactions between the functional groups on the algal surface and MB. After adding algal cells to MB, the solution color became dark blue and gradually disappeared and changed to nearly colorless after the first 24 and 84 h, as shown in Figure 2. Hence, MB decolorization may be due to both the sorption and degradation of MB [28].

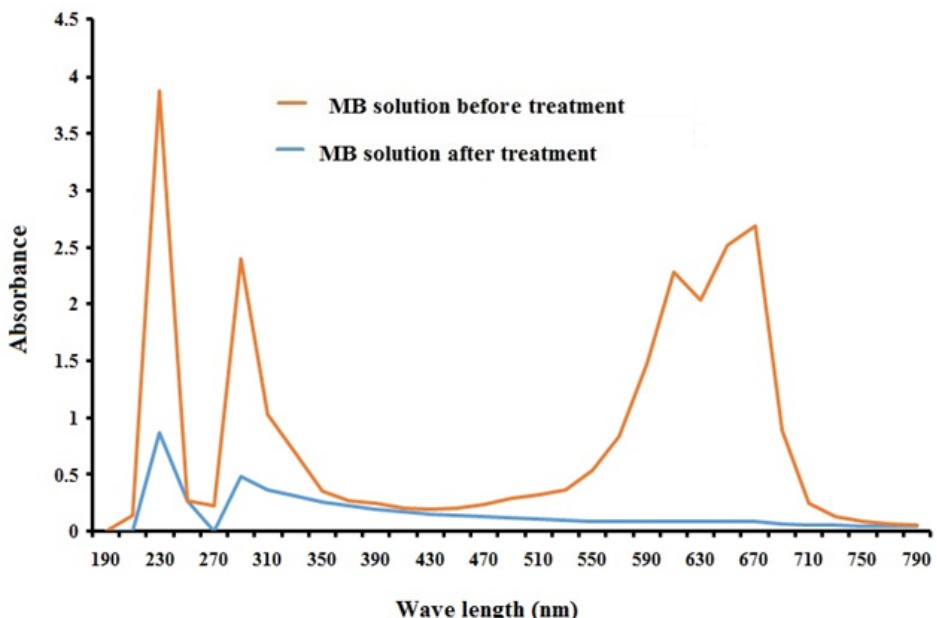

**Figure 1.** UV-Vis absorption spectra of the MB solution before and after treatment with algal biomass after 84h of incubation.

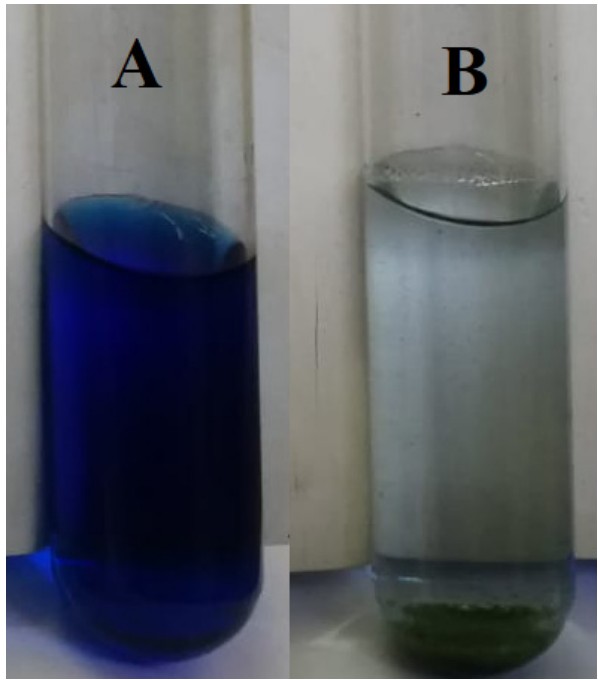

**Figure 2.** Removal of MB using algae biomass. (**A**) MB control and (**B**) MB with algae after 84 h of incubation.

### 3.2. Optimization of the Removal of MB

#### 3.2.1. Effects of Incubation Temperatures, Initial MB Concentrations, and Contact Time

Optimizing the experimental parameters, such as initial MB concentration, incubation temperature, contact time, adsorbent dose, and immobilization, is a key point for obtaining the maximum methylene blue removal from aqueous solutions.

Initial dye concentration, incubation temperature, and contact time are important parameters to determine the optimal removal of methylene blue. Dye removal studies were conducted with different MB concentrations (5–25 mg L$^{-1}$) and three incubation

temperatures (4, 20, and 30 °C) for 84 h (Figure 3), and it was observed that decolorization of MB was found to increase rapidly in the first 24 h. Removal percentages in Figures 3–5 increased as time increased, but removal percentages decreased as methylene blue increased. Figure 3a–c show the effects of incubation temperatures (4, 20, and 30 °C) on the removal percentages of MB; it was observed that as the temperature increased, the removal percentage increased. The best removal percentage was observed atconcentrationsof5 and 10 mg L$^{-1}$and at incubation temperatures of 20 and 30 °C after 84 h.

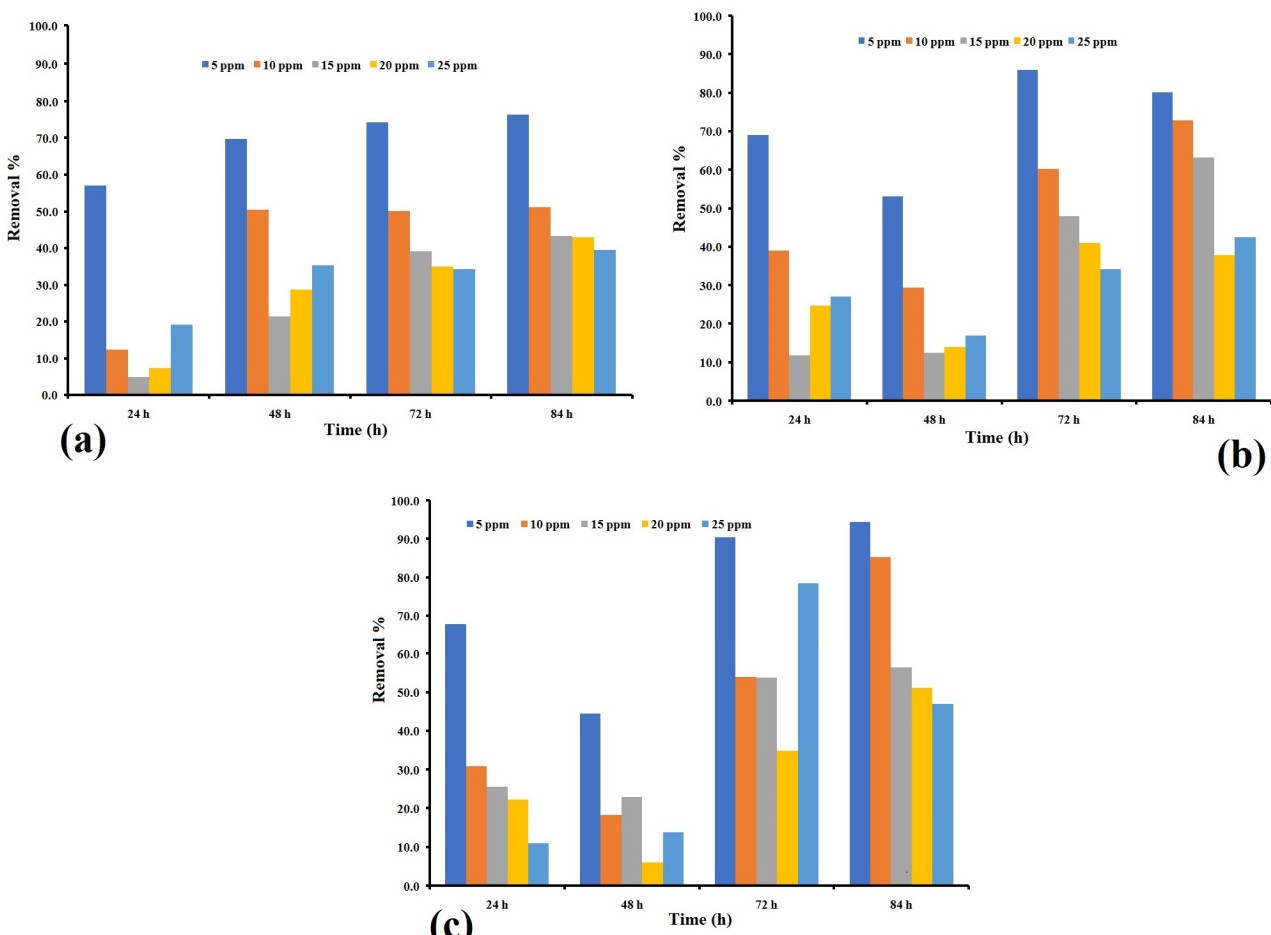

**Figure 3.** (**a**) Effects of initial MB concentration and contact time on MB removal percentage using *Bracteacoccus* sp. (**a**) incubated at 4 °C (**b**) at 20 °C and (**c**) at 30 °C in all temperatures (algae dosage = 0.03 g mL$^{-1}$). All experiments were conducted in triplicate.

*Bracteacoccus* sp. is green microalgae, and its adsorption properties are due to different types of proteins found in the algae cell wall. From the results in Figures 3–5, the contact time shows the gradual effects on the decolorization of MB. This behavior may be because the decolorization process occurs gradually, and the MB molecules need time to be adsorbed into algae sorption sites. The decrease in methylene blue removal happened because of the saturation of the biosorbent site on the cell surface of the algae [29,30]. Temperature is another parameter that affects the decolorization process. The removal percentages increased with increasing temperature from 4 °C to 30 °C due to the increase of sorption sites on the surface of the *Bracteacoccus* sp. cell wall, and these results are in agreement with several previous reports [31]. Several reports have shown that as the temperature increases above 25 °C, the percentage of removal increases as the temperature increases due to the high temperatures inducing dye molecule diffusion in the porous interior structure of the sorbent [32].

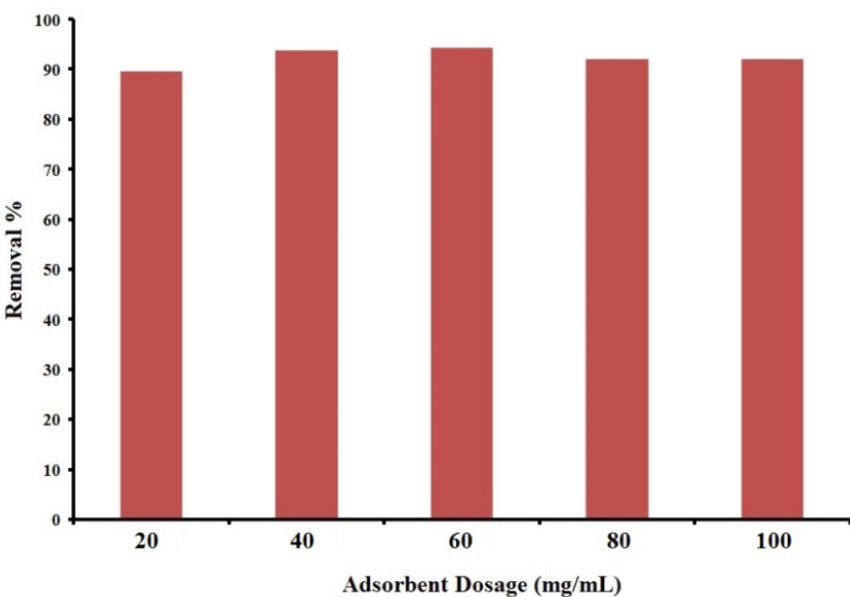

**Figure 4.** Effects of adsorbent dosage on MB ($C_o$ = 15 mg L$^{-1}$) removal efficiency (%) using *Bracteacoccus* sp. incubated at 30 °C for 24 h.

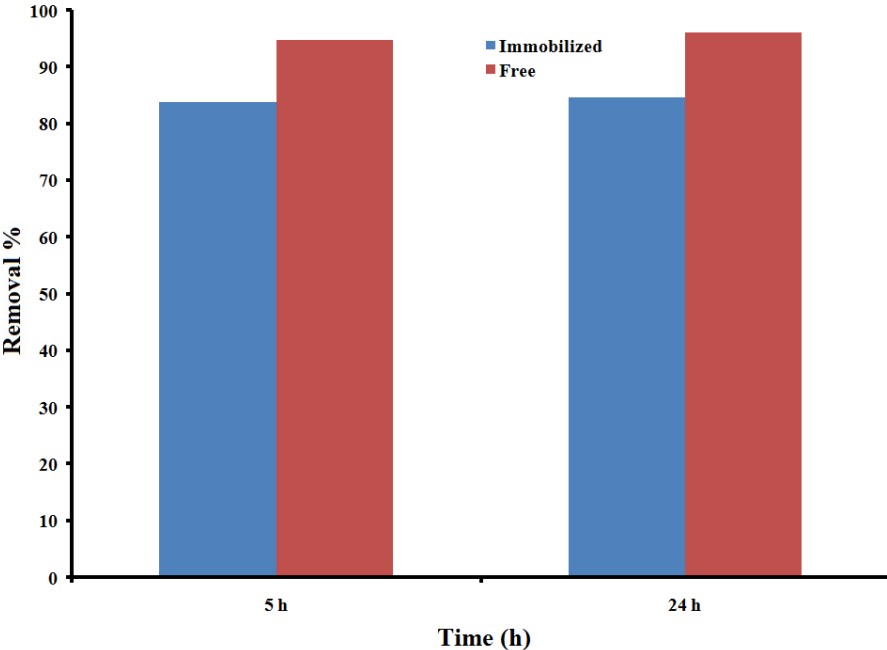

**Figure 5.** Effects of Immobilization on MB removal efficiency (%) ($C_o$ = 15 mg L$^{-1}$, algae dosage = 60 mg/mL, and T = 30 °C after 5 and 24 h).

### 3.2.2. Effects of Adsorbent Dose

Decolorization of methylene blue dye solutions was performed to fix the optimal algal dose required to decolorize MB from the aqueous solutions (Figure 4). From an economic standpoint, determining the amount of adsorbent is very important for selecting the optimum adsorbent dosage for industrial applications [32]. The effects of different algae doses (0.02–0.1 g mL$^{-1}$) are shown in Figure 4. Results showed that when the dosage of algae changed from 0.02 to 0.06 g, the dye removal increased, and the largest quantity of dye removal was attained with adsorbent masses of 0.06 g, with a percentage removal of 94.3%, after that, the removal percentage was decreased (91.9 and 92.0 %) when algae doses were 0.08 and 0.1 g, respectively. Algae dosage affected the decolorization process using

active sites on the algae cell surface and may be attributed to an increase in the biosorption surface area and the number of effective sites on the adsorbent surface. Similar results were reported by other authors [29,31,32]. However, the removal percentages decreased at the higher adsorbent dosage (80 and 100 mg). This may be due to the partial aggregation of *Bracteacoccus* sp. in an aqueous solution, which restricts the number of accessible adsorption sites, or high algal biomass amounts are known to cause aggregation of cells, which will reduce the distance between cells leading to protecting functional groups from MB, as previously reported [32,33].

### 3.2.3. Effects of Immobilization

Most previous reports that used the immobilization of algae focused on the removal of heavy metals; this study supports the use of immobilization for the decolorization process [34]. The percentage of removal in MB dye solution with a 15 mg L$^{-1}$ concentration using immobilized *Bracteacoccus* sp. incubated at 30 °C for 5 and 24 h can be seen in Figure 5. Based on Figure 5, the decolorization percentage of MB by *Bracteacoccus* sp. immobilized in alginate beads at incubation times of 5 h and 24 h, respectively, are 83.8% and 84.7%. The percentage of decolorization of MB by free *Bracteacoccus* sp. at incubation times of 5h and 24h, respectively, are 94.8% and 96.1%. Free and immobilized states of *Bracteacoccus* sp. were able to decolorize methylene blue from aqueous solutions with high capacity reaching 94.8% MB removal. For both incubation times, free *Bracteacoccus* sp. shows the best decolorization efficiency compared with the immobilized state; there was a slight decrease in MB color after 24h when the blank beads (beads without *Bracteacoccus* sp.) were used as biosorbents compared with a clear solution when *Bracteacoccus* sp. was used. Immobilization has many advantages, such as the reusability of cells and its ability for desorption. Immobilization with alginate beads has a semipermeable structure that allows materials with small sizes and water-soluble materials to diffuse in and out of the beads [35]. The removal percentage of free *Bracteacoccus* sp. was slightly higher than immobilized algae beads, and this may be due to the super-concentrated algal cells in alginate beads, which may affect the decolorization process. Immobilization of *Bracteacoccus* sp. in alginate beads is a very efficient method of cell entrapment because immobilization allows high cell loading and light penetration, and it can also be used many times in bioreactors. Alginate beads will protect algal cells from pollutants, such as methylene blue [36].

### Kinetic Models

According to the results from applying pseudo-first-order, pseudo-second-order, and Elovich kinetic models in Equations (1)–(3), the removal of MB using algae is best fitted with the pseudo-second-order model with the highest correlation coefficient ($R^2$ = 0.9997), and the lowest RSME is considered the best fitted for the removal of MB using algae. The parameters and their values related to these kinetic models are listed in Table 1. The order of these kinetic models based on the values of correlation coefficients from high to low are as follows:

$$\text{pseudo-second-order} > \text{Elvoich} > \text{pseudo-first-order}$$

These findings agree with the results of previous studies reported in the literature. According to the values of the correlation factors ($R^2$) listed in Table 1, the pseudo-second-order model has the highest $R^2$ value ($R^2$ = 0.9997), followed by the Elvoich and pseudo-first-order models. Therefore, the pseudo-second-order kinetic model is the best to describe the removal of MB using algae.

### 3.3. Isotherm Models

The isotherm models, such as the Langmuir, Freundlich, Temkin, and D–R, help to explain the nature of the interaction between adsorbate (e.g., MB) and adsorbent surface

(e.g., algae) [37]. The values of parameters related to the isotherm models are listed in Table 2. The order of the isotherm models from high to low $R^2$ is as follows:

$$\text{Langmuir} > \text{Freundlich} \approx \text{D-R} > \text{Temkin}$$

**Table 2.** Langmuir, Freundlich, and Temkin isotherm model constants, correlation coefficients, and error function parameters of MB adsorption on algae.

| Isotherms | Parameters | 4 °C | 20 °C | 30 °C |
|---|---|---|---|---|
| Langmuir | $q_m$ (mg g$^{-1}$) | 0.3603 | 0.4034 | 0.3572 |
| | b (L g$^{-1}$) | 3.0051 | 3.3369 | 2.4986 |
| | $R^2$ | 0.9310 | 0.9413 | 0.9600 |
| Freundlich | $K_F$ ((mg g$^{-1}$) (L mg$^{-1}$)$^{1/n}$) | 0.1477 | 0.1435 | 0.1614 |
| | 1/n | 0.2550 | 0.3055 | 0.2292 |
| | $R^2$ | 0.8766 | 0.8246 | 0.8882 |
| D-R | $q_m$ (mg g$^{-1}$) | 7.2046 | 8.0645 | 6.7935 |
| | $K_D$ (mol$^2$ kJ$^{-2}$) | 3.0043 | 3.3355 | 0.8166 |
| | $\varepsilon$ (KJ mol$^{-1}$) | 0.0624 | 0.0566 | 0.0239 |
| | $R^2$ | 0.8766 | 0.8246 | 0.8882 |
| Temkin | $k_T$ (L mg$^{-1}$) | 1.2055 | 1.5026 | 1.1133 |
| | $b_T$ (kJ mol$^{-1}$) | 1.9115 | 1.6221 | 2.2640 |
| | $R^2$ | 0.8185 | 08031 | 0.8436 |

Consequently, the experimental data are best fitted by the Langmuir model and least fitted by the Temkin model. Thus, a monolayer of MB is formed on the algae surface. This finding agrees with some previous studies [19,38,39].

Comparison between Different Types of Algae

Table 3 shows a comparison between some types of algae used in previous studies as adsorbents for MB removal with the main result obtained from the current study. Among these adsorbents, Immobilized *Bracteacoccus* sp., Algae (*Ulva Lactuca* and *Sargassum*), and the algae used in the current study show the highest removal efficiency of MB (>95%), followed by giant duckweed (*Spirodelapolyrrhiza*). While other bioadsorbents, such as pretreated dead *Streptomyces rimosus* and dead algal biomass, achieve acceptable removal percentages (<70%). Therefore, algae are still effective bioadsorebnets to decolorize contaminated water by MB.

**Table 3.** Maximum uptake capacity of MB on several biosorbents.

| Adsorbent | Experimental Conditions | Removal % $q_{max}$ (mg g$^{-1}$) | Ref. |
|---|---|---|---|
| Marine green alga *Ulva lactuca* | pH = 10, initial MB concentration: 25 g L$^{-1}$, alga concentrations: 2.5 g L$^{-1}$ at room temperature (25 ± 2 °C) | 40.2 mg g$^{-1}$ | [40] |
| Pretreated dead *Streptomyces rimosus* | Biosorbent dose: 5 g L$^{-1}$ Temperature: 20 °C Initial Concentration: 50 mg L$^{-1}$ Contact time: 5 min Sorbent amount: 0.5–2.0 g | 68% removal efficiency (6.93 mg g$^{-1}$) | [41] |
| Green macroalga *Caulerpa lentillifera* | Temperature: 25 °C Initial Concentration: 10 mg L$^{-1}$ Contact time: 1.0 h Sorbent amount: 0.1 g | 417 mg g$^{-1}$ | [38] |
| *Sargassum muticum* | Temperature: room temperature (25 °C). Initial Concentration: 10–1000 mg L$^{-1}$ Contact time: 2.0 h | 279.2 mg g$^{-1}$ | [42] |

**Table 3.** *Cont.*

| Adsorbent | Experimental Conditions | Removal % $q_{max}(mg\ g^{-1})$ | Ref. |
|---|---|---|---|
| Dead macrofungi | Sorbent dose: 0.07–2.10 g L$^{-1}$<br>Temperature: 20 °C<br>Initial Concentration: 100–200 mg L$^{-1}$<br>The pH = 7.5 ± 0.2<br>Contact time: 48 h | 204.38–232.73 mg g$^{-1}$ | [43] |
| Dead algal biomass | T = 20 °C<br>pH = 6.0<br>Initial concentration: 40–800 mg L$^{-1}$<br>Contact time: 3 h | 171, 104, and 74 mg g$^{-1}$, respectively, for algae, algal waste, and composite material (% removals were 48, 32, and 23%, respectively, for algae *Gelidium*, algal waste, and composite material). | [44] |
| Giant duckweed (*Spirodelapolyrrhiza*) | Sorbent amount: 0.1–1.0 g<br>Temperature: 25 ± 2 °C<br>Initial Concentration: 300 mg L$^{-1}$<br>Contact time: 24 h | 250 mg g$^{-1}$(90% removal efficiency) | [45] |
| Brown Alga *Cystoseirabarbatula* Kützing | Temperature:25, 35, and 45 °C. Adsorbent dosage: 0.1 g<br>30 mL of various concentration dye solution (5–100 mg L$^{-1}$)<br>Contact time = 6 h | 38.61 mg g$^{-1}$ at 35 °C | [46] |
| Algae (*Ulva Lactuca* and *Sargassum*) | Co = 3 × 10$^{-6}$–3 × 10$^{-5}$ M<br>Temperature = 303–318 K<br>Adsorbent amount: 0.1–1.0 g<br>Contact time = 25 min | 96% Removal efficiency | [47] |
| *Caulerpa racemosa* var. *cylindracea* | alga amount = 0.1–2.0 g<br>30 mL of the MB solution (5–100 mgL$^{-1}$)<br>pH = 3–11 | 5.23 mg g$^{-1}$ at 18 °C. | [19] |
| Immobilized *Desmodesmus* sp. | Temperature = 27 °C<br>alga dosage = 0.25 g L$^{-1}$<br>pH = 6.8<br>Temperature = 25 °C | 98.6% decolorization efficiency | [48] |
| Green microalgae *Bracteacoccus* sp. | alga dosage = 0.06 g mL$^{-1}$<br>pH = 7.0<br>Temperature = 30 °C, Contact time = 60 min<br>Initial Concentration: 15 mg L$^{-1}$ | 96% removal efficiency (0.3572 mg g$^{-1}$) | current study |

Different low-cost adsorbents (such as *Azolla pinnata* and *GigantochloaBamboo* derived biochar) and their adsorption capacity of methylene blue dye from aqueous solutions were discussed in previous studies, which showed high adsorption performance [49,50].

## 4. Conclusions

Algae are effective biosorbents for MB removal from aqueous solutions. In addition, algae have clear advantages, such as fast growth, availability, high efficiency for MB removal, low cost, and environment friendliness, which make the use of algae a promising route to protect aquatic environments from pollution by industrial effluents, including dyes. Optimization of the experimental parameters, such as initial MB concentration, algae dosage, temperature, and contact time, is an essential step in MB removal from an aqueous solution to obtain the maximum removal efficiency of MB. Based on the decolorization of MB using algae, a potential mechanism of MB removal may include sorption and degradation. Therefore, further studies, including analysis of algae with MB mixture, will be interesting to exactly depict the removal mechanisms of MB using algae.

**Author Contributions:** Conceptualization, A.T.A.-F. and E.E.; data curation, A.A.S., A.T.A.-F. and E.E.; formal analysis, A.T.A.-F. and A.A.S.; methodology, A.T.A.-F., A.A.S. and E.E.; resources, A.T.A.-F., E.E. and A.A.S.; writing—original draft, A.T.A.-F.; writing—review and editing, A.T.A.-F. and E.E. All authors have read and agreed to the published version of the manuscript.

**Funding:** This research received no external funding.

**Institutional Review Board Statement:** Not applicable.

**Informed Consent Statement:** Not applicable.

**Data Availability Statement:** Not applicable.

**Acknowledgments:** The authors would like to thank the Department of Biological Sciences, Al al-Bayt University, for providing administrative and research support.

**Conflicts of Interest:** The authors declare no conflict of interest.

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
