# Peer review of "Bioremoval of Methylene Blue from Aqueous Solutions by Green Algae (Bracteacoccus sp.) Isolated from North Jordan: Optimization, Kinetic, and Isotherm Studies"

_sustainability, doi:10.3390/su15010842_

Round 1

Reviewer 1 Report

In the manuscript, the authors reported the removal of MB from an aqueous solution by Green Algae (Bracteacoccus sp.) Isolated from North Jordan. The article topic is intriguing and promising in the area. Overall, the article structure and content are suitable for the journal Sustainability. I am pleased to send you major-level comments, there are some flaws that need to be corrected before publication.

Please consider these suggestions as listed below.

1. Authors should carefully read the manuscript and correct a lot of typing mistakes.

2. The abstract seems to be fine. Please add one more introductory line of your objective at beginning of the abstract. Highlight the core idea.

3. Authors should remove the ``dyes`` from keywords and put the keywords which are more suitable to the manuscript.

4. In the introduction section, authors should first better describe the environmental problems and the necessity of MB removal.

5. The introduction is well-structured, clear, and easy to follow, but it lacks more information about published works on bio removal. It also needs a better justification for this green algae choice.

6. In subsection 2.1., authors should name all chemicals which are used for conducted research.

7. Authors should revise all equations to the template

8. In figure 1, the authors should improve the plots of the UV-Vis absorbance curves.

9. In the results section, obtained results are too general commented, authors should provide more in-depth discussion.

10. The stability and sensitivity of green microalgae to the targeted pollutant must be discussed.

11. Reference formatting needs careful revision. All must be consistent in one format. Please follow the journal guidelines.

Author Response

We are really thankful for giving us the opportunity to revise our manuscript entitled above. we carefully considered the reviewers’ comments. we want to extend my appreciation for taking the time and effort necessary to provide such insightful guidance. The revision, based on the review team’s collective input, includes a number of positive changes. Based on your guidance, we have accordingly modified the manuscript and detailed corrections, changes and/or rebuttals against each point raised are listed below and addressed in the revised version by red color. Also, we have carefully edited the manuscript and corrected typos and grammatical mistakes. we hope that these revisions improve the paper such that you and the reviewers now deem it worthy for publication.

Reviewer 2 Report

Summary and general comments

This study is interesting to read. The work involves isolating a green algae species, Bracteacoccus, and using it to make adsorbent alginate beads for the removal of methylene blue dye.

Overall, the work is alright and scientifically sound. It is also easy to follow. The selected methodology is suitable for adsorption science. The result and discussion also support the conclusion.

However, the introduction is a little short, and some areas in the methodology sections can be improved as it was not clear. Some figures should be combined, and some sentences should be rewritten. Please see the specific comments below for details.

Specific comments

1.      The introduction is a little short. Authors should describe more about Bracteacoccus so that reader does not need to refer to an external source. Authors should also describe the attractive feature of this algae as an adsorbent.

2.     Authors should briefly discuss major choices of an adsorbent such as biochar or biomass of high availability. Here are some of my recommendations: doi.org/10.1155/2022/8245797 doi.org/10.1016/j.jtice.2021.11.001

3.      There are two sections 2.3

4.      Information about MB dye does not fit in the first Section 2.3. I suggest the author move to section 2.1.

5.      2nd section 2.3a. Is there any study on the shelf life of these algae-adsorbent and whether the adsorption capacity can be affected if these algae is no longer viable?

6.      It will be interesting if the authors show a photograph of the adsorbent beads.

7.      Line 118. Is there any reason to incubate in the dark while algae is more active in the presence of light?

8.      I suggest the author use qe for equation 2 instead of qt. I also recommend the authors to use Ce instead of Ct because Ce is used in the isotherm equations (Eq 6-9).

9.      Authors do not need to define qe for every isotherm equation.

10.   Section 3.1. There is a lot of word redundancy in the first paragraph. The author can straight away mention “The UV-vis spectra….”. There is no need to mention UV-vis spectroscopy used for determining the UV-vis spectra because it is understood

11.   Figures 3,4,5 can be combined as Figure 3a,b,c.

Author Response

(The authors gave the same response as above.)

Round 2

Reviewer 1 Report

The authors have satisfactorily addressed my comments on the previous version of this manuscript. In my opinion, the manuscript has been improved substantially and is acceptable for publication in Sustainability Journal.

Reviewer 2 Report

After the revision, the quality of the manuscript has improved and i recommend it for publication in MDPI sustainability.